# Association of comorbidity between *Opisthorchis viverrini* infection and diabetes mellitus in the development of cholangiocarcinoma among a high-risk population, northeastern Thailand

Kavin Thinkhamrop[1,2,3,4], Narong Khuntikeo[1,2,5], Wongsa Laohasiriwong[6], Pornpimon Chupanit[6], Matthew Kelly[7], Apiporn T. Suwannatrai[1,2,8]*

1 Cholangiocarcinoma Screening and Care Program (CASCAP), Faculty of Medicine, Khon Kaen University, Khon Kaen, Thailand, 2 Cholangiocarcinoma Research Institute (CARI), Khon Kaen, Thailand, 3 Data Management and Statistical Analysis Center (DAMASAC), Faculty of Public Health, Khon Kaen University, Khon Kaen, Thailand, 4 Health and Epidemiology Geoinformatics Research (HEGER), Faculty of Public Health, Khon Kaen University, Khon Kaen, Thailand, 5 Department of Surgery, Faculty of Medicine, Khon Kaen University, Khon Kaen, Thailand, 6 Department of Public Health Administration, Health Promotion, and Nutrition, Faculty of Public Health, Khon Kaen University, Khon Kaen, Thailand, 7 Department of Global Health, Research School of Population Health, Australian National University, Canberra, Australia, 8 Department of Parasitology, Faculty of Medicine, Khon Kaen University, Khon Kaen, Thailand

* apiporn@kku.ac.th

## Abstract

### Background

Cholangiocarcinoma (CCA) is a category of lethal hepatobiliary malignancies. Previous studies have found that *Opisthorchis viverrini* infection and diabetes mellitus (DM) are closely correlated with CCA. However, few studies have discussed the association of CCA with a combination of both *O. viverrini* infection and DM. This study aimed to assess the correlation of CCA with various combinations of *O. viverrini* infection and DM among a high-risk population in northeastern Thailand.

### Methodology

This study included participants from 20 provinces in northeastern Thailand who had been screened for CCA in the Cholangiocarcinoma Screening and Care Program (CASCAP) between 2013 and 2019. Histories of *O. viverrini* infection and DM diagnosis were obtained using a health questionnaire. CCA screening used ultrasonography with a definitive diagnosis based on histopathology. Multilevel mixed-effects logistic regression was performed to quantify the association, which is presented as adjusted odds ratios (aOR) and their 95% confidence intervals (CI).

### Principal findings

Overall, 263,776 participants were included, of whom 32.4% were infected with *O. viverrini*, 8.2% were diagnosed with DM, and 2.9% had a history of both *O. viverrini* infection and DM.

**Data Availability Statement:** The data cannot be shared publicly as it is personal information that must be approved by the committee of the Cholangiocarcinoma Screening and Care Program (CASCAP), Thailand (https://cloud.cascap.in.th/ or cascapkku@gmail.com).

**Funding:** This research was supported by the Young Researcher Development Project of Khon Kaen University and the National Research Council of Thailand (https://www.nrct.go.th/) (grant number RGNS 63 - 049) to KT. This research was also supported by NSRF under the Basic Research Fund of Khon Kaen University through Cholangiocarcinoma Research Institute. The funders had no role in study design, data collection, and analysis, decision to publish, or preparation of the manuscript.

**Competing interests:** The authors have declared that no competing interests exist.

The overall rate of CCA was 0.36%. Of those infected with *O. viverrini*, 0.47% had CCA; among those with DM, 0.59% had CCA and among those infected with *O. viverrini* and had DM, 0.73% had CCA. Compared with participants who were not infected with *O. viverrini* and were non-DM, the aOR for those infected with *O. viverrini* and with DM was 2.36 (95% CI: 1.74–3.21; p-value <0.001).

## Conclusions

The combination of *O. viverrini* infection and DM was highly associated with CCA, and these two conditions had a combined effect on this association that was greater than that of either alone. These findings suggest that CCA screening should have a strong focus on people with a combination of *O. viverrini* infection and DM.

## Author summary

Northeastern Thailand has a high prevalence of liver fluke infection and increasing incidence of diabetes mellitus. These two conditions are individually risk factors for cholangiocarcinoma in at-risk populations. This study used data from the Cholangiocarcinoma Screening and Care Program to assess association of cholangiocarcinoma (CCA) with combinations of *O. viverrini* infection history and diabetes mellitus diagnosis in the high-risk area of northeastern Thailand. We found that infection with *O. viverrini* and a diagnosis of diabetes mellitus (DM) was highly associated with CCA. The risk of CCA in individuals with both of these conditions was higher than in individuals with only one of them.

## Introduction

The liver fluke, *Opisthorchis viverrini* is a food-borne trematode which is a key public health problem with a well-documented distribution in Thailand, Lao PDR, Cambodia, Myanmar, and Vietnam [1,2]. One of the most serious consequences of this liver fluke infection is its association with subsequent cholangiocarcinoma (CCA). Globally the highest prevalence of *O. viverrini* infection, and the highest incidence rates of CCA, are found in Thailand, particularly northeastern Thailand [3–6], where *O. viverrini* infection prevalence was estimated at 17% in 2009 [2]. Another study in 2014 reported the prevalence of *O. viverrini* infection was about 23%, and it was more common in males and among people aged 40–49 years [7].

The main source of infection with *O. viverrini* is the consumption of undercooked or fermented fish (freshwater cyprinid fish) [2,8–10]. This habit is deeply embedded in the food culture of northeastern Thailand, as well as the broader lower Mekong region [11]. Re-infection with *O. viverrini* following curative treatment occurs in around 10% of cases due to this behavior [12]. Continued consumption of raw fish will lead to individuals experiencing cycles of *O. viverrini* infection, treatment, and re-infection, a serious problem in highly endemic areas and one that leads to increased risk of progression to development of CCA [3–6].

*O. viverrini* is not the only risk factor for CCA. Several other diseases have been linked with the condition, including diabetes mellitus (DM) [13,14]. Recent studies have reported that DM is a risk factor for both intrahepatic and extrahepatic subtypes of CCA [15]: patients with DM were 1.6 times more likely to be incident cases of CCA compared with those without DM [16].

In addition, a retrospective study has shown that, although not statistically significant, DM is associated with shorter survival of CCA patients [17]. The relationship between CCA and a combination of *O. viverrini* infection and DM has also been investigated in hamsters, revealing that infection with liver fluke during diabetes leads to more serious disease in the liver than is due to either condition alone [18]. However, there has been no previous investigation in humans of any relationship between CCA and a combination of *O. viverrini* infection and DM. This is particularly important in northeastern Thailand as the region, as well as having high *O. viverrini* infection rates, is also experiencing increasing prevalence of DM. Among rural females in the northeastern part of the country, one study found a prevalence of 8.5% [19]. In addition, the National Health Examination Survey reported an increasing prevalence of DM in the Thai population from 7.7% in 2004 to 9.9% in 2014 [20].

In northeastern Thailand a large-scale CCA screening project is underway run by the Cholangiocarcinoma Screening and Care Program (CASCAP). In this study, residents of northeastern Thailand provide information on both *O. viverrini* infection history and any previous DM diagnosis. They are then screened for CCA and associated hepatobiliary abnormalities. In this study we used CASCAP data to assess associations between the risk of CCA development and *O. viverrini* infection history among people with and without DM. We hypothesize that the group who were infected with *O. viverrini* and who also had a positive DM diagnosis will have a higher risk of CCA than other groups.

## Materials and methods

### Ethics statement

The research protocol was approved by Khon Kaen University Ethics Committee for Human Research, reference number HE631061. The data were provided from the Cholangiocarcinoma Screening and Care Program (CASCAP). The CASCAP data collection was conducted according to the principles of Good Clinical Practice, the Declaration of Helsinki, and national laws and regulations about clinical studies. It was approved by the Khon Kaen University Ethics Committee for Human Research under the reference number HE551404. All patients gave written informed consent for the study.

### Study design

This cross-sectional study collected data from the Cholangiocarcinoma Screening and Care Program (CASCAP), northeastern Thailand. CASCAP is the first project for CCA screening in a high-risk population with a community-based bottom-up approach [21]. The CASCAP screening program aims to recruit all residents of northeastern Thailand. This is achieved using multiple methods and settings including tertiary-care hospitals, district-level hospitals and through mobile screening sessions at the sub-district level. This study includes all participants who were screened for CCA and enrolled in the CASCAP database up to the end of 2019 (n = 263,776). In addition to being screened for CCA, participants also filled out a questionnaire containing socio-demographic information, history of *O. viverrini* infection, diagnosis with DM, as well as other health and lifestyle information.

### Study setting and population

Northeastern Thailand (or Isan) is Thailand's largest region comprising 20 provinces. It is bordered by the Mekong River and Laos to the East and Cambodia to the Southeast. The population of northeastern Thailand is approximately 21 million (around one-third of the total population of Thailand).

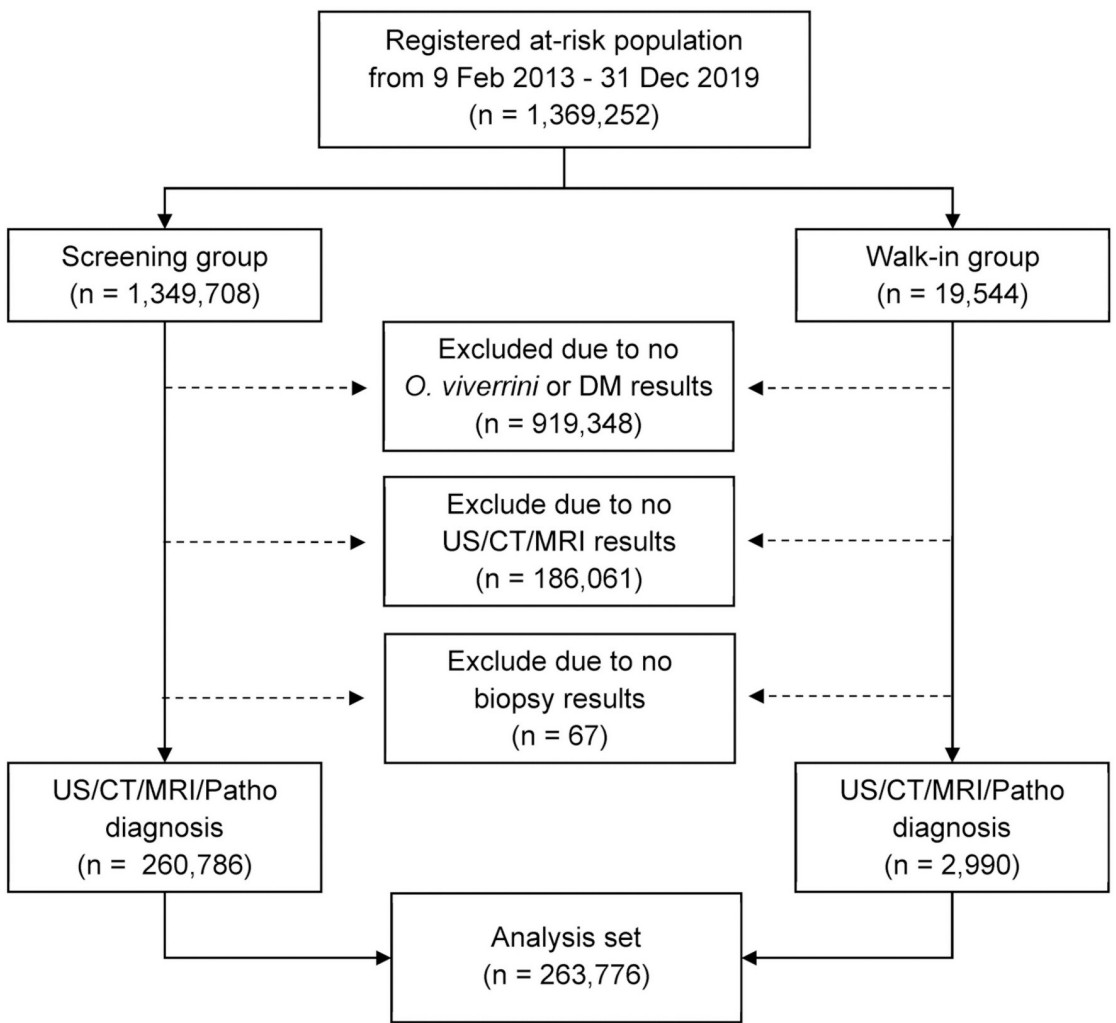

**Fig 1. Data selection process.** Flowchart showing the number of participants at each stage of data selection, leading to the final analysis set.

Our study includes all people who participated in CASCAP from 20 provinces separated into two groups (screening and walk-in). The screening group were people attending health facilities for reasons unrelated to symptoms that could be associated with CCA, and who were invited to undergo routine ultrasonography (US) screening for CCA. The walk-in group comprised people who attended the hospital with symptoms indicating CCA and underwent histo-pathological diagnosis for CCA. Both screening and walk-in groups provided information regarding history of *O. viverrini* infection and DM diagnosis. The sample included in our study were those enrolled in the CASCAP database from 2013–2019 with a total of 263,776 subjects (Fig 1).

## Study outcome and independent variables

The primary outcome for this study was CCA which was defined as histologically positive diagnosis and categorized into two groups (no/yes). CCA was diagnosed through US, CT/ MRI, and histopathological examination. Positive US results were defined as those indicating

liver mass and/or bile duct dilatation, so-called suspected CCA. Those included in this group were referred to receive either CT or MRI scans. For the results of CT/MRI, there were only two categories—positive or negative for CCA. Biopsy confirmation for positive CCA was then needed. Participants who did not have suspicious US nor CT/MRI results were classified as negative for CCA. Each US examination was performed by either radiologists or well-trained general practitioners and was verified centrally by a radiologist at a tertiary hospital. The CT/MRI was mainly done by radiologists at tertiary hospitals. The data of histological findings were based on the standard protocol of the tertiary hospital at Khon Kaen University, Thailand.

The independent variables include the combinations of lifetime history of *O. viverrini* infection and diagnosis of DM. These were categorized into four groups, namely, uninfected with *O. viverrini* and being non-DM (OV- & DM-), uninfected with *O. viverrini* and having DM (OV- & DM+), infected with *O. viverrini* and being non-DM (OV+ & DM-), and infected with *O. viverrini* and having DM (OV+ & DM+). The co-variates include gender, age at enrollment, highest achieved educational level, occupation, history of smoking cigarettes defined as no/yes (current orever), drinking alcohol defined as no/yes (current or previous), history of eating raw fish defined as no/yes (current or previous), and history of praziquantel treatments (PZQ) defined as never/once/twice/three times or more. PZQ is used to treat *O. viverrini* infections. The level factors comprise individual, and province factors.

### Statistical analysis

Categorical demographic characteristics were summarized using frequencies and percentages. Continuous data were summarized by their mean, standard deviation (SD), and minimum and maximum values. The rate of CCA was calculated based on a normal approximation to a binomial distribution. A multilevel mixed-effects logistic regression model was applied to consider the hierarchical structure of the subjects, where the individual (level 1) was nested within the provinces (level 2). The level of association between the combination of *O. viverrini* infection and DM diagnosis adjusted for all other co-variates (fixed effects) and CCA were presented as adjusted odds ratio (aOR) and their 95% confidence intervals (CI). This model was adjusted for the variation of individual-level, and province-level effects (random effects). The highest value of maximum likelihood was estimated to assess the fitness of the model. All test statistics were two-tailed and a p-value of less than 0.05 was considered statistically significant. All analyses were performed using a statistical package, STATA version 15 (Stata, College Station, Texas, USA).

## Results

### Descriptive summary

A total of 263,776 participants who underwent screening for CCA were enrolled in our study (Table 1). Participants were aged from 18 to 110 years with a mean age of 55.7 (SD = 9.3) years. Almost two-thirds of them were female (60.7%). The majority had only completed primary school or had not completed any formal education (76%) and worked as farmers (84.3%). Among study participants, overall prevalence of *O. viverrini* infection was 32.4%, and overall prevalence of DM was 8.2%. Among the participants, 62.3% (164,258) were in the OV- & DM- group, 5.4% (14,163) were OV- & DM+, 29.5% (77,831) were OV+ & DM-, and 2.8% (7,524) were in the OV+ & DM+ group. Fig 2 shows the gender distribution of each of these groups. Females outnumbered males in the OV+ and DM+ group. Fig 3 The highest percentage of participants who had OV+ and DM+ were in the >60 years category at 4.4% (3,483/ 78,625).

**Table 1. Demographic characteristics of participants according to *O. viverrini* infection and diabetes mellitus status.** The data are presented as frequencies and percentages for the overall sample and separated by *O. viverrini* infection and diabetes mellitus status.

| Characteristics | Total | Not infected with *O. viverrini* (OV-) | | Infected with *O. viverrini* (OV+) | |
|---|---|---|---|---|---|
| | | DM- | DM+ | DM- | DM+ |
| | n = 263,776 (%) | n = 164,258 (%) | n = 14,163 (%) | n = 77,831 (%) | n = 7,524 (%) |
| Gender | | | | | |
| Female | 160,128 (60.7) | 101,698 (61.9) | 10,014 (70.7) | 43,402 (55.8) | 5,014 (66.6) |
| Male | 103,641 (39.3) | 62,556 (38.1) | 4,149 (29.3) | 34,426 (44.2) | 2,510 (33.4) |
| Age groups (years) | | | | | |
| < 50 | 72,549 (27.7) | 50,257 (30.8) | 1,834 (13.1) | 19,639 (25.4) | 819 (10.9) |
| 50–60 | 110,774 (42.3) | 68,333 (41.9) | 6,019 (42.8) | 33,261 (43.0) | 3,161 (42.4) |
| > 60 | 78,625 (30.0) | 44,528 (27.3) | 6,201 (44.1) | 24,413 (31.6) | 3,483 (46.7) |
| Mean (SD) | 55.7 (9.3) | 54.9 (9.3) | 59.2 (8.4) | 56.2 (9.2) | 59.7 (8.1) |
| Educational levels | | | | | |
| Primary and lower | 200,418 (76.0) | 121,550 (74.0) | 11,608 (81.9) | 60,996 (78.4) | 6,264 (83.2) |
| Secondary | 51,554 (19.5) | 34,664 (21.1) | 1,994 (14.1) | 13,919 (17.9) | 977 (13.0) |
| Certificate and higher | 11,804 (4.5) | 8,044 (4.9) | 561 (4.0) | 2,916 (3.7) | 283 (3.8) |
| Occupation | | | | | |
| Unemployed | 8,841 (3.4) | 5,120 (3.1) | 1,080 (7.6) | 2,179 (2.8) | 462 (6.2) |
| Farmer | 222,444 (84.3) | 137,590 (83.8) | 11,280 (79.7) | 67,341 (86.5) | 6,233 (82.8) |
| Others | 32,491 (12.3) | 21,548 (13.1) | 1,803 (12.7) | 8,311 (10.7) | 829 (11.0) |
| Smoking history | | | | | |
| No | 204,765 (77.6) | 129,475 (78.8) | 11,762 (83.1) | 57,465 (73.8) | 6,063 (80.6) |
| Yes, current or previous | 59,011 (22.4) | 34,783 (21.2) | 2,401 (16.9) | 20,366 (26.2) | 1,461 (19.4) |
| Alcohol consumption | | | | | |
| No | 145,099 (55.0) | 91,097 (55.5) | 8,960 (63.3) | 40,327 (51.8) | 4,715 (62.7) |
| Yes, current or previous | 118,677 (45.0) | 73,161 (44.5) | 5,203 (36.7) | 37,504 (48.2) | 2,809 (37.3) |
| History of raw fish eating | | | | | |
| No | 21,651 (8.2) | 15,513 (9.4) | 1,058 (7.5) | 4,644 (6.0) | 436 (5.8) |
| Yes, current or previous | 242,125 (91.8) | 148,745 (90.6) | 13,105 (92.5) | 73,187 (94.0) | 7,088 (94.2) |
| Praziquantel treatments | | | | | |
| Never | 137,570 (52.1) | 120,459 (73.3) | 10,576 (74.7) | 6,069 (7.8) | 466 (6.2) |
| Once | 96,982 (36.8) | 30,361 (18.5) | 2,388 (16.9) | 58,654 (75.3) | 5,579 (74.2) |
| Twice | 18,887 (7.2) | 7,622 (4.6) | 642 (4.5) | 9,544 (12.3) | 1,079 (14.3) |
| Three times and more | 10,337 (3.9) | 5,816 (3.6) | 557 (3.9) | 3,564 (4.6) | 400 (5.3) |

Abbreviations: SD, Standard deviation; n, Number of participants; DM, Diabetes mellitus

## Rate and association summary

Table 2 reveals the associations between frequency of *O. viverrini* infection, DM status and CCA diagnosis. From a total of 263,776 participants, 0.36% were diagnosed with CCA. The prevalence rate of CCA was 0.47% for those who had ever been infected with *O. viverrini*, 0.59% for those with DM. Compared to participants who were not infected with *O. viverrini*, the aOR for CCA among those infected with *O. viverrini* was 1.63 (95% CI: 1.37–1.92; p-value <0.001). Compared to participants who did not have DM, the AOR for CCA among those with DM was 1.50 (95% CI: 1.24–1.82; p-value <0.001). These effects controlled for co-variates comprising gender, age at enrollment, educational levels, occupation, smoking cigarettes, drinking alcohol, eating raw fish, and PZQ treatments (fixed effect), and the variation of province levels (random effects).

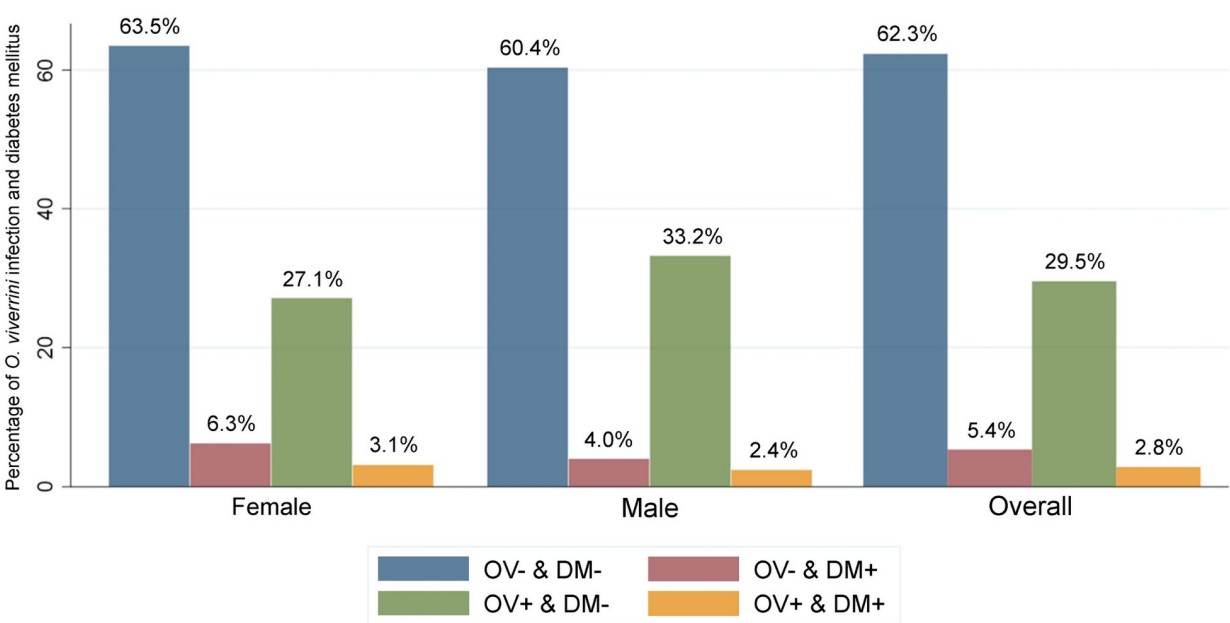

**Fig 2. Breakdown of study population according to *O. viverrini* infection and diabetes mellitus status.** Data show percentages of the study population falling into each of the four possible combinations of *O. viverrini* infection and diabetes mellitus status according to gender and overall.

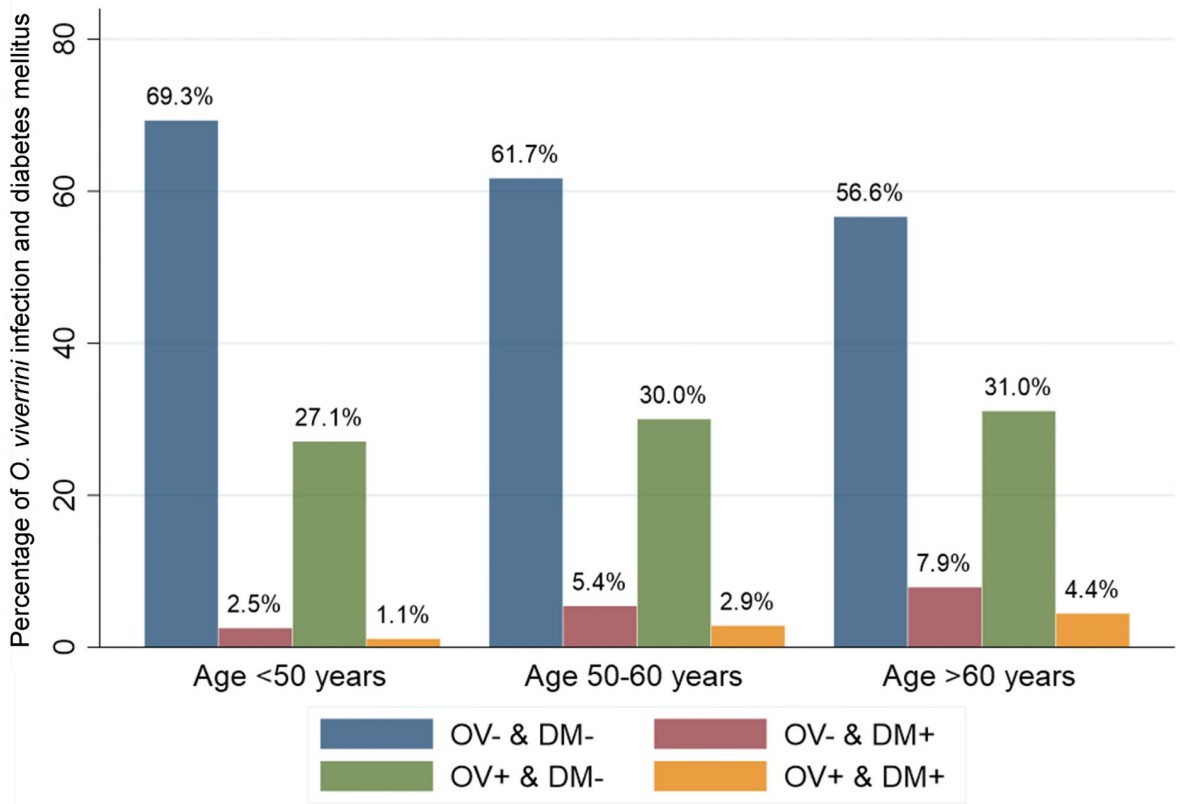

**Fig 3. Breakdown of study population according to *O. viverrini* infection and diabetes mellitus status.** Data show percentages of the study population falling into each of the four possible combinations of *O. viverrini* infection and diabetes mellitus status according to age groups.

**Table 2. Association of *O. viverrini* infection, diabetes mellitus diagnosis and other factors with cholangiocarcinoma using multilevel mixed-effects logistic regression.** The data are presented as number of participants, number and percentage having cholangiocarcinoma, crude odds ratios with their 95% confidence intervals and p-values from likelihood-ratio chi-square tests, and adjusted odds ratios and their 95% confidence intervals and p-values from likelihood-ratio chi-square tests for each factor.

| Factors | Number | CCA | | Crude analysis | | | Adjusted analysis | | |
|---|---|---|---|---|---|---|---|---|---|
| | | Cases | % | cOR | 95% CI | *P-value* | aOR | 95% CI | *P-value* |
| Overall | 263,776 | 944 | 0.36 | NA | NA | NA | NA | NA | NA |
| *O. viverrini* infection | | | | | | <0.001 | | | <0.001 |
| No | 178,421 | 544 | 0.30 | 1 | | | 1 | | |
| Yes | 85,355 | 400 | 0.47 | 1.61 | 1.41–1.83 | | 1.63 | 1.37–1.92 | |
| Diabetes mellitus | | | | | | <0.001 | | | <0.001 |
| No | 242,089 | 815 | 0.34 | 1 | | | 1 | | |
| Yes | 21,687 | 129 | 0.59 | 1.71 | 1.42–2.06 | | 1.50 | 1.24–1.82 | |
| Gender | | | | | | <0.001 | | | 0.039 |
| Female | 160,128 | 336 | 0.21 | 1 | | | 1 | | |
| Male | 103,641 | 608 | 0.59 | 2.77 | 2.42–3.17 | | 0.81 | 0.67–0.99 | |
| Age groups (years) | | | | | | <0.001 | | | <0.001 |
| < 50 | 72,549 | 87 | 0.12 | 1 | | | 1 | | |
| 50–60 | 110,774 | 335 | 0.30 | 2.42 | 1.91–3.06 | | 2.14 | 1.69–2.72 | |
| > 60 | 78,625 | 518 | 0.66 | 5.25 | 4.18–6.60 | | 4.68 | 3.69–5.93 | |
| Educational levels | | | | | | | | | <0.001 |
| Primary and lower | 200,418 | 734 | 0.37 | 1 | | <0.001 | 1 | | |
| Secondary | 51,554 | 109 | 0.21 | 0.55 | 0.45–0.67 | | 0.66 | 0.54–0.82 | |
| Certificate and higher | 11,804 | 101 | 0.86 | 2.30 | 1.86–2.83 | | 1.67 | 1.30–2.15 | |
| Occupation | | | | | | | | | <0.001 |
| Unemployed | 8,841 | 44 | 0.50 | 1 | | <0.001 | 1 | | |
| Farmer | 222,444 | 685 | 0.31 | 0.64 | 0.47–0.87 | | 0.82 | 0.60–1.12 | |
| Others | 32,491 | 215 | 0.66 | 1.33 | 0.96–1.85 | | 1.71 | 1.21–2.42 | |
| Smoking history | | | | | | <0.001 | | | <0.001 |
| No | 204,765 | 419 | 0.20 | 1 | | | 1 | | |
| Yes, current or previous | 59,011 | 525 | 0.89 | 4.32 | 3.80–4.92 | | 2.69 | 2.23–3.25 | |
| Alcohol consumption | | | | | | <0.001 | | | <0.001 |
| No | 145,099 | 239 | 0.16 | 1 | | | 1 | | |
| Yes, current or previous | 118,677 | 705 | 0.59 | 3.70 | 3.20–4.29 | | 2.35 | 1.97–2.81 | |
| History of raw fish eating | | | | | | <0.001 | | | <0.001 |
| No | 21,651 | 31 | 0.14 | 1 | | | 1 | | |
| Yes, current or previous | 242,125 | 913 | 0.38 | 2.86 | 1.99–4.09 | | 2.01 | 1.40–2.90 | |
| Praziquantel treatment | | | | | | <0.001 | | | <0.001 |
| Never | 137,570 | 401 | 0.29 | 1 | | | 1 | | |
| Once | 96,982 | 257 | 0.26 | 0.87 | 0.74–1.02 | | 0.58 | 0.48–0.71 | |
| Twice | 18,887 | 153 | 0.81 | 2.43 | 2.02–2.94 | | 1.51 | 1.21–1.87 | |
| Three times and more | 10,337 | 133 | 1.29 | 3.70 | 3.03–4.53 | | 2.27 | 1.83–2.83 | |

Abbreviations: NA, Not applicable; cOR, Crude odds ratio; aOR, Adjusted odds ratio; 95% CI, 95% confidence interval of adjusted odds ratio

The combination of *O. viverrini* infection and DM, was associated with the highest rate of CCA, 0.73% (55/7,524). The highest CCA rate was 1.35% (34/2,510) found in males in the OV+ & DM+ group (Fig 4). Fig 5 shows the number of CCA cases according to the combination of *O. viverrini* infection and DM and separated by age.

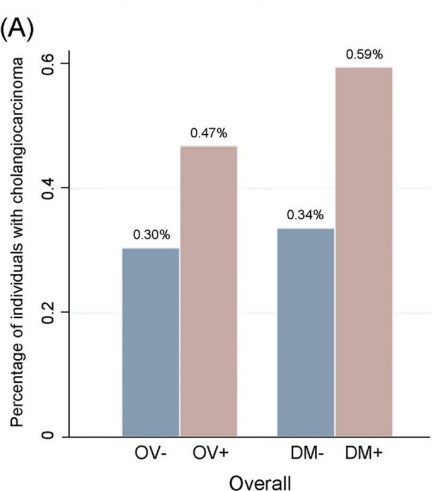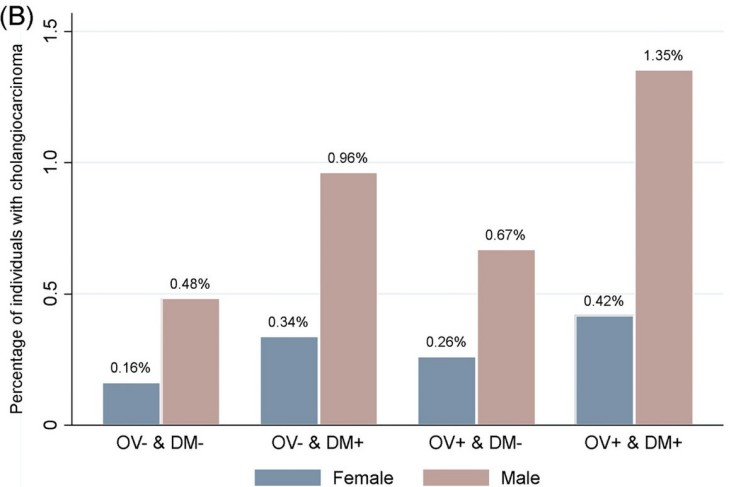

**Fig 4. Percentages of individuals with cholangiocarcinoma in relation to *O. viverrini* infection and diabetes mellitus status overall (A) and according to gender (B).** Data show the rate of cholangiocarcinoma as percentage for overall *O. viverrini* infection and diabetes mellitus groups, and combination of *O. viverrini* infection and diabetes mellitus separated by sex.

Multilevel mixed-effects logistic-regression model results for the association of combinations of *O. viverrini* infection and DM diagnosis with CCA are reported in Table 3 and Fig 6. Compared to participants in the OV- & DM- group, the crude analysis showed an OR of 2.5 (95% CI: 1.88–3.31; p-value <0.001) for CCA among those in the OV+ & DM+ group. In multivariable analysis, the aOR for CCA among those in the OV+ & DM+ group was 2.36 (95%

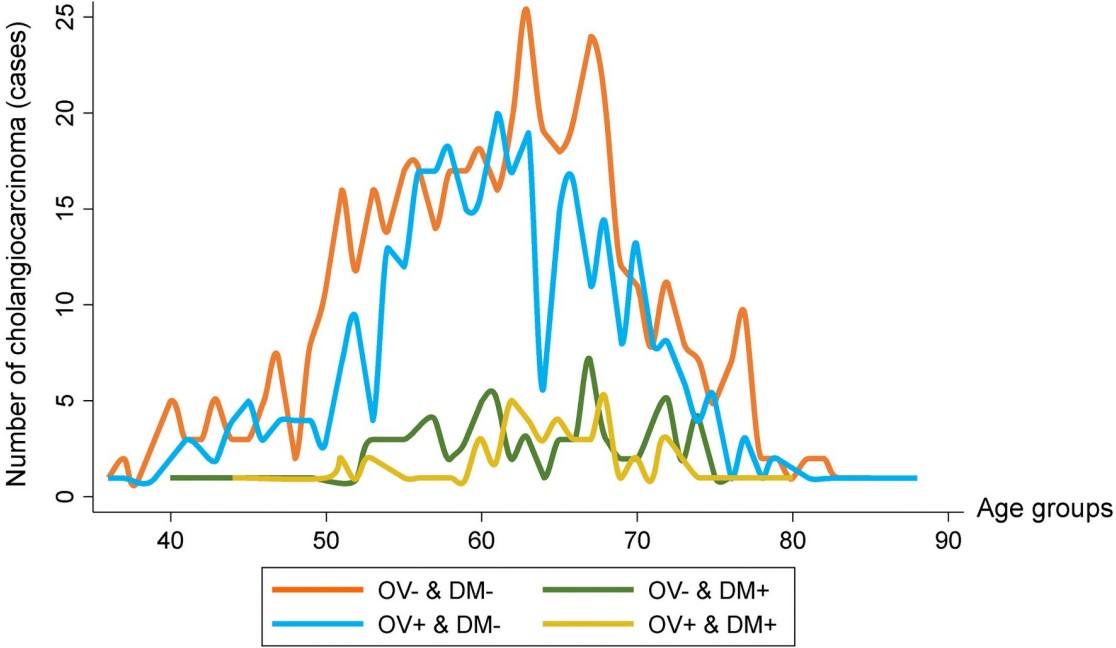

**Fig 5. Numbers of cholangiocarcinoma cases by age and in relation to *O. viverrini* infection and diabetes mellitus status.** Data shows the number of cholangiocarcinoma cases according to combination of *O. viverrini* infection and diabetes mellitus according to age in year.

**Table 3. Association of combinations of *O. viverrini* infection and diabetes mellitus with cholangiocarcinoma using multilevel mixed-effects logistic regression.**
The data are presented as numbers of participants, numbers and percentages having cholangiocarcinoma, crude odds ratios and their 95% confidence interval and p-value from likelihood-ratio chi-square tests, and adjusted odds ratios and their 95% confidence interval and p-values from likelihood-ratio chi-square tests for various combinations of *O. viverrini* infection and diabetes mellitus.

| Factors | Number | CCA | | Crude analysis | | | Adjusted analysis | | |
|---|---|---|---|---|---|---|---|---|---|
| | | Cases | % | cOR | 95% CI | *P-value* | aOR | 95% CI | *P-value* |
| Combination of *O. viverrini* infection and diabetes mellitus | | | | | | | | | |
| OV- & DM- | 164,258 | 470 | 0.29 | 1 | | <0.001 | 1 | | <0.001 |
| OV- & DM+ | 14,163 | 74 | 0.52 | 1.81 | 1.41–2.31 | | 1.56 | 1.21–2.00 | |
| OV+ & DM- | 77,831 | 345 | 0.44 | 1.63 | 1.42–1.88 | | 1.64 | 1.38–1.96 | |
| OV+ & DM+ | 7,524 | 55 | 0.73 | 2.50 | 1.88–3.31 | | 2.36 | 1.74–3.21 | |

Abbreviations: OV, *Opisthorchis viverrini*; DM, Diabetes mellitus; CCA, Cholangiocarcinoma; cOR, Crude odds ratio; aOR, Odds ratio adjusted for gender, age, educational levels, occupation, smoking cigarettes, drinking alcohol, eating raw fish, and praziquantel treatment; 95% CI, 95% confidence intervals

CI: 1.74–3.21; p-value <0.001). These effects were controlled for co-variates comprising gender, age at enrollment, educational levels, occupation, smoking cigarettes, drinking alcohol, eating raw fish, PZQ treatments (fixed effects), and the variation of province level (random effects).

## Discussion

We investigated the association of CCA with various combinations of DM and *O. viverrini* infection in Northeast Thailand. Our study revealed that CCA was highly associated with the combination of *O. viverrini* infection and DM and that this association was higher than that of

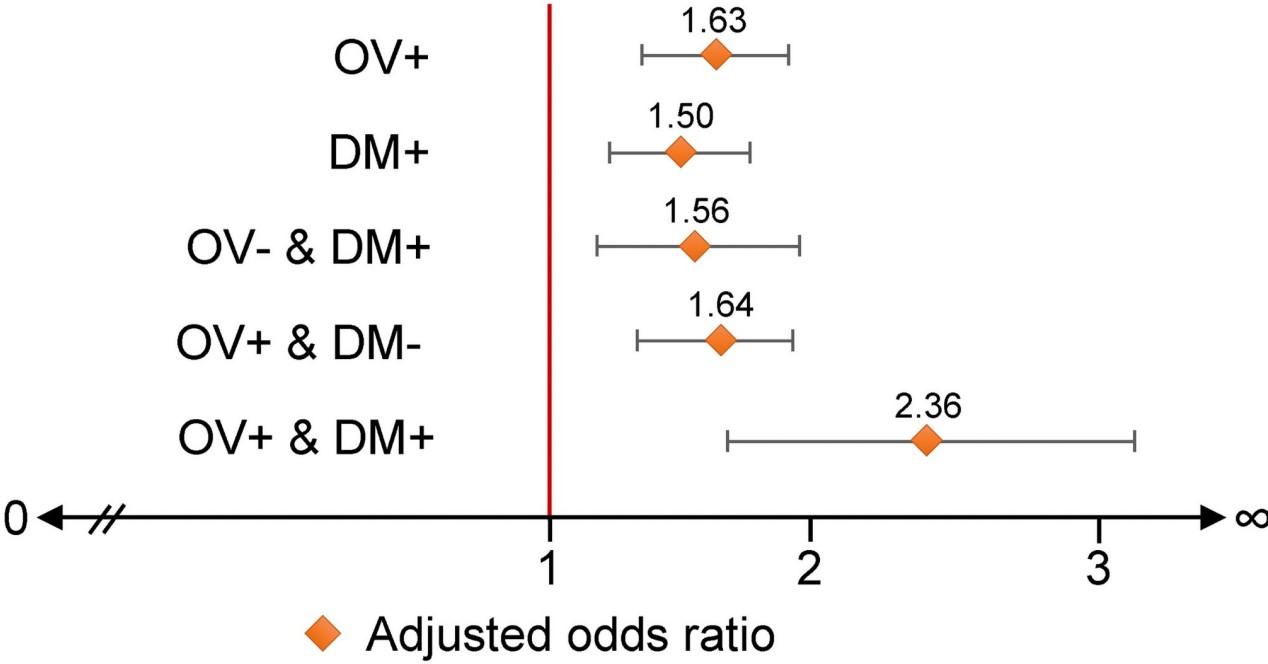

**Fig 6. Adjusted odds ratios for the association of cholangiocarcinoma with *O. viverrini* infection and diabetes mellitus status.** Data show the magnitude of association of cholangiocarcinoma, comparing groups infected with *O. viverrini* (OV+), diabetes mellitus (DM+), and combinations of these (OV- & DM+, OV+ & DM- and OV+ & DM+). In each case, the comparison is against the group without diabetes mellitus or *O. viverrini* infection.

CCA with either *O. viverrini* infection alone or DM alone. To the best of our knowledge, these findings represent the first epidemiological observation among a large population in Northeast Thailand concerning co-occurrence of DM and *O. viverrini*, and the association with CCA risk.

The proportion of our participants (32.4%) who had a history of *O. viverrini* infection was higher than noted in 2014, in a study which reported a prevalence of about 23% across a few provinces in northeastern Thailand [7]. This is likely because CASCAP is a large screening program for CCA that covers the entire region. We also found that 21,687 participants (8.2%) had DM, which is close to the previously reported prevalence in northeastern Thailand of about 8.5% [19]. The overall rate of CCA in our study was 0.36%. Of those infected with *O. viverrini*, 0.47% had CCA; among those with DM, 0.59% had CCA and among those in the OV + & DM+ group, 0.73% had CCA. Given the nature of the study, where CCA case rates were calculated from among those who underwent screening, it is difficult to compare these results with other studies. However, these results are concerning given that northeastern Thailand is estimated to contribute 63% of CCA cases nationally, while containing only about one-third of the population of Thailand [22].

We found that participants who had been infected with *O. viverrini* had a 63% higher chance of having CCA than those who were not infected. Our study was conducted in an area endemic for *O. viverrini* infection, a major risk factor for development of CCA [23–26]. In line with other previous reports [13–15], our study also found that the chance of having CCA increased by 50% for people who had DM. Our study area, besides being a liver-fluke endemic area, also has an increasing incidence of DM [19,20]. Epidemiological studies are increasingly indicating DM as a risk factor for CCA [27–32]. DM plays a key role in influencing carcinogenesis and promoting progression of CCA [28].

Our findings showed CCA had the greatest association with the OV+ & DM+ group in both crude and multivariable analysis, and that the association was larger than for the *O. viverrini* infection group or the DM group alone. This is consistent with a previous study conducted in a hamster model [18]. Potential mechanisms to explain this increased CCA risk have been explored in some studies. For example, *O. viverrini* infection worsens diabetes-related damage to the liver and bile ducts in rodents, CCA patients' tissues, and cultured human CCA cells [18,33]. As well, co-occurrence of opisthorchiasis and DM in hamsters and tissue cultures of human liver cells has been found to lead to more severe disease in the liver than is caused by either condition alone [18]. High glucose levels can stimulate the proliferation and metastatic ability of CCA cells derived from patients with chronic *O. viverrini* infections [33].

Other studies have assessed whether DM is a risk factor or protective factor in patients diagnosed with biliary tract diseases. One study in a non-endemic area for liver-fluke infection reported that DM may be a protective factor against CCA development in those with biliary tract diseases. However, DM was also associated with significantly increased risk of CCA in patients without these conditions [14]. This finding is consistent with another study in northeastern Thailand, which found that, before praziquantel (PQZ) treatment, *O. viverrini* infection had a protective effect against hyperglycemia and risk of metabolic disease. However, serum levels of HbA1c and HDL significantly increased during the six months following PQZ treatment [34]. In addition, a study among high-risk subjects for CCA who were diagnosed with periductal fibrosis (PDF) by ultrasonography found that DM had a protective effect against PDF [35]. The association between DM and *O. viverrini*-associated CCA needs further investigation both in molecular and epidemiological studies. There is still some possibility that the increased risks for CCA development that are associated with infection by *O. viverrini* coexisting with DM may be coincidental rather than causative.

In conclusion, our findings have revealed that *O. viverrini* infection and DM diagnosis are highly associated with CCA, with those positive for both *O. viverrini* infection and DM being at the highest risk. These findings suggest that CCA-screening programs, as well as health-education efforts, should concentrate not only on *O. viverrini* infection and DM separately but also should have a particular focus on those with a combination of these two risk factors.

## Limitations of the study

A limitation of our study was that the data regarding history of *O. viverrini* infection and DM diagnosis were self-reported by participants, resulting in potential recall bias in the study results. This may particularly affect reporting of *O. viverrini* infections by older participants in whom infection may have occurred a long time prior to study participation. Detailed dietary data were not collected. All participants reported a history of previously consuming raw/fermented fish but information on frequency, interval and amount of consumption was not assessed. This information may have given more explanatory power to the differences in CCA observed between groups of study participants.

## Recommendations

This study was conducted in northeastern Thailand and may not reflect the general population of the country or of SE Asia more generally. Further study is necessary in the region to test the generality of our results. Nevertheless, the methodology and results of our study can be used as a guideline in formulating clinical practice and future research priorities.

## Acknowledgments

The authors truly thankful for all members of CASCAP, particularly the cohort members and staff from all participating institutions including the Ministry of Public Health, Ministry of Interior, and Ministry of Education of Thailand. We would like to acknowledgement Prof. David Blair, for editing the MS via Publication Clinic KKU, Thailand.

## Author Contributions

**Conceptualization:** Kavin Thinkhamrop, Apiporn T. Suwannatrai.

**Data curation:** Kavin Thinkhamrop, Narong Khuntikeo, Apiporn T. Suwannatrai.

**Formal analysis:** Kavin Thinkhamrop, Apiporn T. Suwannatrai.

**Investigation:** Kavin Thinkhamrop, Narong Khuntikeo, Wongsa Laohasiriwong, Pornpimon Chupanit, Matthew Kelly, Apiporn T. Suwannatrai.

**Methodology:** Kavin Thinkhamrop, Apiporn T. Suwannatrai.

**Project administration:** Kavin Thinkhamrop, Narong Khuntikeo, Apiporn T. Suwannatrai.

**Resources:** Kavin Thinkhamrop, Narong Khuntikeo.

**Supervision:** Kavin Thinkhamrop, Narong Khuntikeo, Wongsa Laohasiriwong, Apiporn T. Suwannatrai.

**Validation:** Kavin Thinkhamrop, Narong Khuntikeo, Wongsa Laohasiriwong, Pornpimon Chupanit, Matthew Kelly, Apiporn T. Suwannatrai.

**Writing – original draft:** Kavin Thinkhamrop, Apiporn T. Suwannatrai.

**Writing – review & editing:** Kavin Thinkhamrop, Narong Khuntikeo, Wongsa Laohasiriwong, Pornpimon Chupanit, Matthew Kelly, Apiporn T. Suwannatrai.

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
