## [Decision Letter · Decision Letter 0]

9 May 2021

Dear Dr. Suwannatrai,

Thank you very much for submitting your manuscript "Association of comorbidity between liver fluke infection and diabetes mellitus in the development of cholangiocarcinoma among high risk population, Northeast of Thailand" for consideration at PLOS Neglected Tropical Diseases. As with all papers reviewed by the journal, your manuscript was reviewed by members of the editorial board and by several independent reviewers. The reviewers appreciated the attention to an important topic. Based on the reviews, we are likely to accept this manuscript for publication, providing that you modify the manuscript according to the review recommendations. 

Sincerely,

Sutas Suttiprapa, Ph.D.

Associate Editor

Banchob Sripa

Deputy Editor

Reviewer's Responses to Questions

**Key Review Criteria Required for Acceptance?**

**Methods**

-Are the objectives of the study clearly articulated with a clear testable hypothesis stated?

-Is the study design appropriate to address the stated objectives?

-Is the population clearly described and appropriate for the hypothesis being tested?

-Is the sample size sufficient to ensure adequate power to address the hypothesis being tested?

-Were correct statistical analysis used to support conclusions?

-Are there concerns about ethical or regulatory requirements being met?

Reviewer #1: Feasible

Reviewer #2: = objective clearly stated

= study design is appropriate

= Hypothesis missing. proposed to clearly spell-out the hypothesis in the last paragraph of the instroduction

= sample size is fixed as CASCAP data was used

= statistical analysis should be reviewed by statistician

= yes, data originates from large-scale ongoing screening program. Ethics statement is not given in manuscript. Should be added.

please address the following important points:

= please provide for all variables used in this analysis a clear definition:

1. outcome: please state whether CCA was defined as histologically positive CCA or also other assessments.

2. O. viverrini infection: state clearly that this is a reported history of O. viverrini infection. Was the question restricted to a time period or any prior diagnosis.

3. Diabetes assessment was also reported by the patient. What exactly was asked. Ever diagnosed? Has the current treatment been included

4. smoking: please clarify if current smoking, ever smoking, smoking quantity etc.

5. drinking alcohol: please clarify what is considered drinking alcohol? is linked with quantity etc. or provide question.

6. eating raw fish: please specify what exactly was asked to participant (ever eaten raw fish, fish sauce, fermented fish etc.)

= please improve the statement of the statistical analysis in order to be:

When performing the adjusted analysis, did the authors adjust always for all the other variable. In Table 2 is it correct that for each aOR provided the 9 other variables were used for adjustment. Or, did the author conduct one multivariable analysis and report the adjusted results?

**Results**

-Does the analysis presented match the analysis plan?

-Are the results clearly and completely presented?

-Are the figures (Tables, Images) of sufficient quality for clarity?

Reviewer #1: Clear presentation

Reviewer #2: = results presented match the analysis plan

= result presentation can be improved:

1. In descriptive summary

- please add also overall prevalence of Opisthorchis infection, DM and CCA.

2. Table 1

- please add in all columns a column-percent in brackets as it is done in the "Total" column. This will add more numbers to the table but provides the reader an easier insight into the data which currently difficult because the reported number of observations in the subgroups are huge.

3. Table 2

This is the key results table of the manuscript. Hence it is worthy to add elements to provide easier insights in the data (landscape presentation might be necessary). Suggested to:

- add column percent in the "number" column

- insert a column next to "number" column (left of "%CCA" column) in which absolute number of CCA cases are reported

- insert two column in which 95% Ci and p-value of cOR are reported

- insert a column in which 95% Ci and p-value of aOR are reported

- please clarify the role of the reported p-value

4. Table 3

- please modify the table in analogy of table 2

- the content of this table is largely represented by Fig 6. Therefore, table 3 could be moved to a supplementary file

5. Fig 1

- can the exact dates of used registration period be given (1 Jan 2013 - 31 Dec 2019)?

6. Fig 2

- please use also colors.

- please add overall section with prevalence of O. viverrini infection and DM

7. Fig 3

- please use also colors

8. Fig 4

- please use also colors

- suggested to add overall section with CCA incidence in Ov alone and DM alone

9. Fig 5

- please use also colors

- suggestion: it might be clearer with the smoothed age-prevalence curves

10. Fig 6

- very nice! It could replace Table 3, see above

- please provide exact aOR above the rhomboid symbol

11. All tables and figures

- please provide in addition to the title a descriptive text with details in order to make the table/figure self-explanatory

**Conclusions**

-Are the conclusions supported by the data presented?

-Are the limitations of analysis clearly described?

-Do the authors discuss how these data can be helpful to advance our understanding of the topic under study?

-Is public health relevance addressed?

Reviewer #1: Somewhat weak

Reviewer #2: The authors report on a CCA risk increase when Ov infection and DM is present and was highest when both exposure were present. However, the discussion regarding the internal validity of the study is rather weak. The recall bias is mentioned in the limitations but could be addressed more clearly. Further other biais as seletion bias is likely to present. In Fig 1 clearly shows that almost 1 million people were excluded from the analysis. Could they have resulted in a selection biais. In addition, study participants consisted of "screening" and "walk-in" group patient groups. In how far could this fact have influenced the observed associations? Finally, existing evidence on the underlying biological mechanisms is given in the introduction. However, it would be great if this section in the introduction (lines 118-125) could be moved to the discussion and could be elaborated on in more detail. It is interesting that one cited article (Tsai et al, Int J Cancer, 2015; ref 14) reports on a negative association between DM and CCA.

Consequently, the conclusion of the article could be adapted to this discussion part and tuned down a bit. However, the public health message is nicely stated and more attention should be given to the multi-morbidity groups.

**Editorial and Data Presentation Modifications?**

Reviewer #1: (No Response)

Reviewer #2: = Please, carefully English edit the text. E.g. "had O. viverrini positive" or "had DM positive" is not correct English; line 101: drop "old", line 133-4: suggest to revise sentence; line 173: pathological or histological? line 163-4: Ov infection prevalences are higher in Laos than in NE Thailand; reported CCA incidence are worldwide highest in NE Thailand;

= Please check references and edit according to the journals recommendations. Italicize genus/species names.

= please check sentence on line 105: meaning seems inadequate. Most Ov re-infection are due to raw/fermented fish consumption

**Summary and General Comments**

Reviewer #1: General Comments

This is a large scale cross-sectional retrospective study on the effects of co-morbidity of Ov infection and DM on cholangiocarcinogenesis.

The story is smiple and the experimental design is straightforward. The results are clear-cut.

In contrast, the MS is lengthy, wordy with many repetitions. Introduction and Discussion can be shortened considerably to make MS much more impressive.

Specific Comments

Although this MS has a co-author of native-English using scientist, there are numerous typos and English usage problems, although those does not affect the scientific quality of this work.

1) Typically, for study areas, “Northeast of Thailand” in the title, but “Northeastern Thailand”, “Northeast Thailand”, “Thailand’s Northeast region” in other places. I believe the Thai government’s official expression is “the Northeast Thailand” and “northeastern Thailand”. Consistency is required.

2) Opishorchis viverrini is the name of parasite, whereas Diabetes mellitus is the name of disease. Whenever you are talking about co-morbidity, or combination of O. viverrini infection and DM, you should use “opisthorchiasis” or “O. viverrini infection” instead of “O. viverrini” 

Discussion section is too much repeat of the results. What is important is how DM and Ov infection mutually affect to lead carcinogenesis. Please modify discussion to provide more about possible mechanisms of co-morbidity of opisthorchiasis and DM on CCA genesis.

Reviewer #2: Congratulations! This is a extremely nice and most valuable report. It is based on a unique data base, CASCAP. The conclusions on the study are of importance for Southeast Asia, particularly because of O. viverrini rates are high in many places and diabetes prevalence is in the increase.

PLOS authors have the option to publish the peer review history of their article (what does this mean?). If published, this will include your full peer review and any attached files.

Reviewer #1: No

Reviewer #2: No

Figure Files:

Data Requirements:

Reproducibility:

References

---

## [Decision Letter · Decision Letter 1]

6 Jul 2021

Dear Dr. Suwannatrai,

Thank you very much for submitting your manuscript "Association of comorbidity between Opisthorchis viverrini infection and diabetes mellitus in the development of cholangiocarcinoma among a high risk population, northeastern Thailand" for consideration at PLOS Neglected Tropical Diseases. As with all papers reviewed by the journal, your manuscript was reviewed by members of the editorial board and by several independent reviewers. The reviewers appreciated the attention to an important topic. Based on the reviews, we are likely to accept this manuscript for publication, providing that you modify the manuscript according to the review recommendations, especially the English usage. 

Sincerely,

Sutas Suttiprapa, Ph.D.

Associate Editor

Banchob Sripa

Deputy Editor

Reviewer's Responses to Questions

**Key Review Criteria Required for Acceptance?**

**Methods**

-Are the objectives of the study clearly articulated with a clear testable hypothesis stated?

-Is the study design appropriate to address the stated objectives?

-Is the population clearly described and appropriate for the hypothesis being tested?

-Is the sample size sufficient to ensure adequate power to address the hypothesis being tested?

-Were correct statistical analysis used to support conclusions?

-Are there concerns about ethical or regulatory requirements being met?

Reviewer #1: (No Response)

Reviewer #2: please see general comments

**Results**

-Does the analysis presented match the analysis plan?

-Are the results clearly and completely presented?

-Are the figures (Tables, Images) of sufficient quality for clarity?

Reviewer #1: (No Response)

Reviewer #2: Authors have improved the presentation of the results. In the result tables of the risk analysis p-values are provided. Please specify from which statistics the p-values are (from the LR chi2 or z-statistics?). Please specify in the table legend. If the p-value is from z-statistics then it should be given on the same row as the OR and the 95% CI of the OR.

The Figures and Tables have been improved and are now self-explanatory. However, the explanatory text need to be English edited.

**Conclusions**

-Are the conclusions supported by the data presented?

-Are the limitations of analysis clearly described?

-Do the authors discuss how these data can be helpful to advance our understanding of the topic under study?

-Is public health relevance addressed?

Reviewer #1: (No Response)

Reviewer #2: good and have been improved

**Editorial and Data Presentation Modifications?**

Reviewer #1: (No Response)

Reviewer #2: It is still necessary that the text is English edited.

Please, check Table 3. The presentation is confusing. Explanations of the adjustments should be given in the legend. The p-value seems to be indicated in the wrong row. The crude and adjusted OR should be always corrected abbreviated (aOR, cOR) and the abbreviations consistently used (in this table and in the entire manuscript). Please also introduce the aOR and cOR in the analysis section of the methods.

**Summary and General Comments**

Reviewer #1: Comments to PNTD-D-21-00483R1

This revised version is acceptable after some corrections for English usages.

Especially attention is required for the usage of the abbreviations like OV and DM for the group names.

Consistent use of abbreviation is required.

l. 89: were >> are

l.94: people >> those

l.103: delete “and” after Thailand

l.107: delete “in Thailand”

l.110: by >> with

l.120: in >> of

l.119-120: change to “has shown that, although statistically not significant, DM was associated with shorter survival of CCA patients [17].

l.121: reported >> revealed

l.123: delete “the”

l.135: underway as the

l.148: requested from >> provided from

l.193: This group were >> Those included in this group were

l.196: negative results >> negative results group

l.199: data on >> data of

l.202: diagnosis with DM >> diagnosis of having DM

l.202: in to >> into

In the l.201-205 in the M&M section, the authors have defined 4 groups of participants using abbreviation of OV for O. viverrini infection and DM for diabetes mellitus.

In the Results section, the group names have been used inconsistently. Typical example can be seen in l.237-245, but inconsistency throughout in the Results and Discussion.

Consistent use of abbreviation is required.

Reviewer #2: This is a very interesting study. The authors have addressed all the points raised in my first assessment.

PLOS authors have the option to publish the peer review history of their article (what does this mean?). If published, this will include your full peer review and any attached files.

Reviewer #1: No

Reviewer #2: No

Figure Files:

Data Requirements:

Reproducibility:

References

---

## [Decision Letter · Decision Letter 2]

17 Aug 2021

Dear Dr. Suwannatrai,

We are pleased to inform you that your manuscript 'Association of comorbidity between Opisthorchis viverrini infection and diabetes mellitus in the development of cholangiocarcinoma among a high-risk population, northeastern Thailand' has been provisionally accepted for publication in PLOS Neglected Tropical Diseases.

Best regards,

Sutas Suttiprapa, Ph.D.

Associate Editor

Banchob Sripa

Deputy Editor

Reviewer's Responses to Questions

**Key Review Criteria Required for Acceptance?**

**Methods**

-Are the objectives of the study clearly articulated with a clear testable hypothesis stated?

-Is the study design appropriate to address the stated objectives?

-Is the population clearly described and appropriate for the hypothesis being tested?

-Is the sample size sufficient to ensure adequate power to address the hypothesis being tested?

-Were correct statistical analysis used to support conclusions?

-Are there concerns about ethical or regulatory requirements being met?

Reviewer #1: (No Response)

Reviewer #2: OK

**Results**

-Does the analysis presented match the analysis plan?

-Are the results clearly and completely presented?

-Are the figures (Tables, Images) of sufficient quality for clarity?

Reviewer #1: (No Response)

Reviewer #2: OK

**Conclusions**

-Are the conclusions supported by the data presented?

-Are the limitations of analysis clearly described?

-Do the authors discuss how these data can be helpful to advance our understanding of the topic under study?

-Is public health relevance addressed?

Reviewer #1: (No Response)

Reviewer #2: OK

**Editorial and Data Presentation Modifications?**

Reviewer #1: (No Response)

Reviewer #2: OK

**Summary and General Comments**

Reviewer #1: (No Response)

Reviewer #2: The authors have addressed all comments / suggestions.

PLOS authors have the option to publish the peer review history of their article (what does this mean?). If published, this will include your full peer review and any attached files.

Reviewer #1: No

Reviewer #2: No

---

## [Editor Report · Acceptance letter]

3 Sep 2021

Dear Dr. Suwannatrai,

We are delighted to inform you that your manuscript, "Association of comorbidity between Opisthorchis viverrini infection and diabetes mellitus in the development of cholangiocarcinoma among a high-risk population, northeastern Thailand," has been formally accepted for publication in PLOS Neglected Tropical Diseases.

Best regards,

Shaden Kamhawi

co-Editor-in-Chief

Paul Brindley

co-Editor-in-Chief
